# Proteome-Wide Identification and Comparison of Drug Pockets for Discovering New Drug Indications and Side Effects

**DOI:** 10.3390/molecules30020260

**Published:** 2025-01-10

**Authors:** Renxin Zhang, Zhiyuan Chen, Shuhan Li, Haohao Lv, Jinjun Li, Naixue Yang, Shaoxing Dai

**Affiliations:** 1State Key Laboratory of Primate Biomedical Research, Institute of Primate Translational Medicine, Kunming University of Science and Technology, Kunming 650500, China; zhangrenxin1999@163.com (R.Z.); chenzhiyuan_7715@163.com (Z.C.); lishuhan1@stu.kust.edu.cn (S.L.); lhhmail508@163.com (H.L.); 20222140064@stu.kust.edu.cn (J.L.); 2Yunnan Key Laboratory of Primate Biomedical Research, Kunming 650500, China

**Keywords:** druggability, proteome, drug repurposing, side effects prediction, drug–protein interactions

## Abstract

Drug development faces significant financial and time challenges, highlighting the need for more efficient strategies. This study evaluated the druggability of the entire human proteome using Fpocket. We identified 15,043 druggable pockets in 20,255 predicted protein structures, significantly expanding the estimated druggable proteome from 3000 to over 11,000 proteins. Notably, many druggable pockets were found in less studied proteins, suggesting untapped therapeutic opportunities. The results of a pairwise pocket similarity analysis identified 220,312 similar pocket pairs, with 3241 pairs across different protein families, indicating shared drug-binding potential. In addition, 62,077 significant matches were found between druggable pockets and 1872 known drug pockets, highlighting candidates for drug repositioning. We repositioned progesterone to ADGRD1 for pemphigus and breast cancer, as well as estradiol to ANO2 for shingles and medulloblastoma, which were validated via molecular docking. Off-target effects were analyzed to assess the safety of drugs such as axitinib, linking newly identified targets with known side effects. For axitinib, 127 new targets were identified, and 46 out of 48 documented side effects were linked to these targets. These findings demonstrate the utility of pocket similarity in drug repositioning, target expansion, and improved drug safety evaluation, offering new avenues for the discovery of new indications and side effects of existing drugs.

## 1. Introduction

Drug development has contributed considerably to human health, but the process is fraught with many challenges and difficulties [1]. The creation of new medicines requires a lengthy process and substantial financial investment. The entire process can cost tens of billions of dollars and take 10 to 15 years from the initial discovery of a new compound to the eventual launch of a new drug. Notably, these figures have continued to increase in recent years [2,3,4]. The number of new drugs approved by the U.S. Food and Drug Administration (FDA) has declined, whereas the cost of drug development has continued to rise. Only 4% of drug development projects ultimately result in the approval of a drug [5]. The drug development process is broadly divided into two phases: drug discovery and drug development [6]. The drug discovery phase is further subdivided into target identification and validation, lead screening, and lead optimization [7,8]. Among these steps, molecular target identification and validation are the initial stages of drug development and are of paramount importance. The majority of drug development efforts begin with the identification of a biologically relevant protein target using a variety of biological methods, followed by the search for and validation of suitable small-molecule ligands.

Drug repositioning (new use of old drugs) is emerging as an increasingly attractive complementary strategy, given its relatively low development costs and shortened development timeline as well as the relatively low success rate of new drug development. The indications and side effects of these drugs are ascertained by identifying potential targets for drugs that have not yet been discovered. Liu et al. developed a multitask graph neural network model for identifying the potential targets of drugs based on the chemical structure information of the compound [9]. Sawada et al. employed a method based on the binding affinity of a drug to a protein to predict both the therapeutic effects and the potential side effects [10]. However, studies investigating drug repositioning through the lens of protein pocket similarity are lacking.

In 2002, Hopkins and Groom first introduced the concept of the “druggable genome”, which refers to the subset of proteins expressed by the approximately 30,000 genes in the human genome that can bind to drug-like molecules [11]. Genomic data suggest that at least 3000 genes encode proteins with the potential to become drug targets. However, only 10% of these potential targets have been translated into FDA-approved drugs [11]. Edwards et al. found that the majority of research (75%) currently focuses on just 10% of the known mammalian proteins [12]. The human genome encodes more than 500 protein kinases, a significant proportion of which have been implicated in human disease. Approximately 65% of the 20,000 papers published on kinase-related topics have focused on just the top 50 proteins. A significant number of potential targets have not been sufficiently studied or have not been studied at all [13]. Whether other druggable proteins exist in addition to the 3000 known druggable proteins should be examined. Druggable parts may be hidden in proteins in which researchers have been less interested or have overlooked. These questions remain unanswered.

The druggability of proteins is typically analyzed on a limited number of protein classes. Lewis et al. conducted a druggability analysis and classified the structure of proteins containing the bromine domain, identifying proteins with potential relevance for drug discovery [14]. Similarly, Christine et al. analyzed the druggability of only the potential allosteric sites in kinases [15]. The rarity of comprehensive druggability analyses of the human proteome is largely due to the time-consuming and costly nature of the experimental determination of the three-dimensional structure of proteins. Currently, only 14% and 21% of the sequences of the human proteome have undergone partial and comprehensive three-dimensional structure confirmation, respectively [10]. Notably, 46% of all human proteins are completely homologous and over 90% have partial homology, indicating that the three-dimensional structures of a significant proportion of human proteins can be tentatively determined using prediction methods. The application of artificial intelligence to the prediction of protein three-dimensional structures has recently demonstrated greater accuracy than traditional homology modeling methods (such as MODELLER [16]) and ab initio computational methods (such as ROSETTA [17]), with AlphaFold2 [18] in particular showing exceptional accuracy at CASP14 in 2020 [19]. The comprehensive druggability analysis enabled by AlphaFold’s prediction of the three-dimensional structures of the human proteome is a substantial advance in the field of protein research.

In this study, we used the three-dimensional structures of all human proteins predicted by AlphaFold2 to analyze the druggability of the human proteome, using Fpocket to predict druggable pockets. We identified 15,043 druggable pockets in 20,255 predicted protein structures. We found pairs of similar pockets and repositioned targets based on the similarity comparison between the human proteome pockets and known druggable pockets with the aim of discovering new indications and side effects for drugs.

## 2. Results

### 2.1. Identifying Druggable Pockets from Tdark Proteins and Expanding Horizons in Human Proteome Druggability

We first collected all of the predicted protein structures of the human proteome from AlphaFold2 and analyzed the druggability of the entire proteome by identifying the predicted druggable pockets. The druggability across the human proteome was assessed by quantifying the druggable pockets, which was followed by a comparative analysis with known drug molecule pockets to facilitate drug repurposing and identify potential novel side effects (Figure 1). We predicted the pockets in 20,255 proteins using the Fpocket tool [20] and identified 11,378 druggable proteins containing 15,043 druggable pockets (drug score > 0.5), as shown in Figure 2A. A parallel analysis using P2Rank (v2.4) [21] identified 6113 druggable proteins with 7985 druggable pockets (probability > 0.5) (Appendix A), and using PUResNetV2 (v0.1) [22] identified 11,378 druggable proteins with 14,450 druggable pockets. The intersection between Fpocket, P2Rank, and PUResNetV2 predictions, shown in Appendix A, revealed 4069 druggable proteins identified using all approaches. The number of pockets predicted by PUResNetV2 intersecting with those predicted by Fpocket, shown in Appendix A, indicated that most pockets predicted by PUResNetV2 could be predicted by Fpocket. Our findings exceed the prior estimate of 3000 druggable proteins regardless of the approach, highlighting a broader druggable landscape in the human proteome than previously known.

A further analysis of druggability according to the target development level (TDL) was performed. Briefly, Tclin are druggable targets with approved drugs, Tchem are potent small molecule binders, Tbio are proteins with disease associations but no small molecule binding, and Tdark are proteins that do not meet Tclin, Tchem, or Tbio criteria (see Section 4 for detailed definitions). The well-studied Tclin and Tchem proteins exhibited druggable proportions of 69.47% and 65.12%, respectively (Figure 2B). The less studied or unstudied Tbio and Tdark proteins had druggable rates of 54.60% and 54.84%, respectively. Overall, the proportion of druggable proteins was higher in the well-studied proteins than in the less studied or unstudied proteins. More than 50% of the less studied or unstudied proteins were found to be druggable, indicating the untapped druggable potential of these proteins. Figure 2C illustrates druggability at the protein family level. The GPCR, transporter, and nuclear receptor families showed the highest druggable proportions of 94.44%, 89.96%, and 85.42%, respectively. The “Other” category demonstrated a druggable rate of over 50%, highlighting the significant druggable potential of the proteome. We compared the proportions of druggable proteins predicted by Fpocket and PUResNetV2 for each family, respectively. The differences between the two methods were minimal (Appendix A).

Our focus then shifted to the Tdark proteins, which are largely unstudied and curated only at the sequence level. The top 20 Tdark proteins with the highest druggability, ranked by drug score, are listed in Appendix A. Of these, we highlight one highly druggable protein in each category (other, ion channel, GPCR, and enzyme) in Figure 2D. These proteins possess pockets capable of binding small-molecule ligands, suggesting the presence of numerous druggable targets within the Tdark proteins. These findings reveal markedly broader druggable potential within the human proteome than previously recognized and confirm numerous potential druggable targets in understudied categories such as Tdark proteins. This expanded view not only supports the prospect of drug repurposing, but also opens avenues for the exploration of novel therapeutic targets and potential side effects.

### 2.2. Druggable Pocket Network in the Human Proteome Reveals Pocket Similarity Among Different Protein Families

A total of 15,043 pockets with a drug score greater than 0.5 were initially identified in the human proteome. We filtered and retained 12,350 eligible pockets considering the requirement of at least 10 amino acid residues for Apoc [23] analysis. Apoc was then applied to pairwise-compare these pockets, excluding intraprotein pairs, which resulted in 76,251,204 pocket pairs, of which 220,312 pairs showed significant similarity (*p*-value < 0.0001) (Figure 3A). The results are visualized in a network graph (Appendix A). We screened for 3241 pairs of similar pocket pairs that exist between proteins in different families and with different domains, so these pockets may have greater potential importance in drug repurposing.

A network graph of target families was constructed from the 3241 pocket pairs (Figure 3B), highlighting the prominent families among the top 20 with the highest number of similar interfamily pocket pairs, including the KRAB and PRY families (from the epigenetic family) and the Ion_trans family (ion channel family) (Figure 3C). The KRAB family, one of the largest vertebrate transcriptional regulatory families, acts together with the KAP1 cofactor to repress transposable element derived sequences [24]. The PRY family, associated with the SPRY domain, is involved in various biological processes, such as cell signaling and immune responses [25]. The Ion_trans family, crucial for maintaining intracellular balance and nerve signal transmission, includes TRP channels, which are integral to multiple physiological pathways and represent attractive drug targets due to their distinct crystal structure and therapeutic potential [26].

The PRY/KRAB, KRAB/Ion_trans, and KRAB/RUN pairs were identified as having the largest number of similar pocket pairs upon further comparison of the family pairs (Figure 3D). The RUN family, which interacts with specific GTPases, is critical for regulating intracellular signaling and influencing cellular behavior [27]. Our findings revealed the pocket similarity among different protein families, providing new insights into potential drug targets. Notably, the KRAB protein family forms more similar pockets with several other protein families, suggesting that it possesses a diverse range of binding pocket types. This diversity increases the likelihood that drugs targeting other protein families could be repurposed to target the KRAB protein family. In contrast, the PRY/KRAB family shares the most similar pockets among them, making it more likely to be targeted by a specific drug or class of drugs.

### 2.3. Similarity Network Between Known Druggable Pockets and Our Identified Pockets Reveals Potential Targets for Drug Repositioning

A comprehensive pairwise comparison of 12,350 druggable pockets from the human proteome with 1872 known druggable pockets was performed using the Apoc tool. From this analysis, we identified a total of 23,119,200 potential pocket pairs, of which 62,077 pairs exhibited significant similarity (*p*-value < 0.0001) (Figure 4A). A network graph detailing these similar pocket pairs is shown in Appendix A. A total of 1114 similar pocket pairs were retained for the final analysis after further filtering based on family and domain information.

The network graph of these 1114 pairs illustrates the relationships between drug molecules, known targets, and potential repositioning targets (Figure 4B). Notably, the top 20 protein families with the highest number of similar pocket pairs relative to drug molecule pockets include the Pkinase_C, PRY, and Ant_C families (Figure 4C). This suggests that the proteins within these families are likely candidates for drug repositioning efforts. The Pkinase_C family (protein kinase C terminal domain) includes several subtypes that are classified into three main subfamilies based on their second messenger dependency: cPKC, nPKC, and aPKC, playing critical roles in various cellular processes [28]. The Ant_C family, associated with the C-terminal region of the anthrax toxin receptor, is also critical in the pathogenesis of anthrax disease [29,30].

A further comparative analysis of the family pairs revealed that the Pkinase_C/PSI, PRY/HSP90, and Pkinase_C/Gly_rich pairs have the highest number of similar pocket pairs (Figure 4D). This suggests that Pkinase_C family proteins could be repurposed for drug molecules targeting PSI and Gly_rich, whereas the PRY family proteins may be repositioned by drugs targeting HSP90. The Plexin repeat family, which are receptors for the neural guidance molecule semaphorin, is involved in the development and pathology of multiple systems [31,32]. HSP90 is notable for its role in regulating the cell cycle and signaling pathways, making it a significant target in cancer therapy [33]. The glycine-rich family (Gly_rich) is widespread among eukaryotes, though comprehensive comparative studies are still lacking [34]. Overall, these findings offer valuable insights into the potential for drug repositioning. As with other protein families, those with the greatest number of pockets are more likely to be repositioned using existing drugs (e.g., Pkinase_C). Similarly, when viewed in the context of family pairs, Pkinase_C family proteins are most likely to be repositioned using drugs acting on the PSI family.

### 2.4. Repositioning Drugs for New Indications Through Drug–Protein–Indication Networks

After constructing the network of drug molecules, known targets, and repositioning targets, we associated these targets with their respective indications, providing a novel perspective on drug repositioning. The following examples illustrate this process.

Progesterone, a multifunctional hormone traditionally used for contraception, managing abnormal uterine bleeding, and preventing endometrial hyperplasia, was found to interact with the GPCR target ADGRD1 (Q6QNK2) through repositioning via a nuclear receptor, as shown in Figure 5A. This interaction is linked to various indications, including pemphigus, atopic dermatitis, and breast cancer. Molecular docking studies revealed a binding energy of −6.9 kcal/mol between progesterone and ADGRD1, indicating a significant interaction.

Figure 5B presents estradiol, an estrogen steroid commonly used to treat menopausal symptoms and estrogen deficiency and to prevent osteoporosis. Through repositioning via a nuclear receptor, estradiol was found to interact with the ion channel target ANO2 (Q9NQ90), which is associated with indications such as shingles and medulloblastoma. The molecular docking results showed a binding energy of −7.3 kcal/mol, suggesting a strong interaction between estradiol and ANO2.

Figure 5C shows alprazolam, a triazolobenzodiazepine used primarily for treating anxiety disorders. By repositioning through an epigenetic target, alprazolam was found to interact with the “Other” category target DNAH1 (Q9P2D7), which is related to indications such as Pick’s disease and nasopharyngeal carcinoma. The results of molecular docking analysis indicated a binding energy of −6.2 kcal/mol, demonstrating an effective interaction.

By mapping these drug interactions to new disease indications, this drug–protein–indication network identified previously unrecognized therapeutic potentials for known drugs. For example, progesterone has been identified as a potential therapeutic agent for diseases such as pemphigus through the repositioning of GPCR targets. These findings reveal new therapeutic potential for known drugs and provide valuable information for the development of new indications for drugs.

### 2.5. Identifying Drug Side Effects Through Drug–Protein–Side Effect Networks

We constructed a drug–protein–side effect network to understand the side effects of drugs by linking drug molecules to both known and repositioned targets and assessing their potential relationships with the observed side effects. This approach enables the exploration of side effects beyond conventional on-target mechanisms, providing deeper insights into drug safety.

Axitinib was found to have 99 known targets and 4 known pockets. We successfully identified 127 targets for axitinib by analyzing pocket similarity, as detailed in Table 1. Of these targets, 24 are identical to previously known targets, validating the methodology used. Notably, 46 of the 48 documented side effects of axitinib are associated with these newly predicted targets. Serious side effects such as hypertension, thrombosis, bleeding, and gastrointestinal perforation are associated with 103 of the newly identified targets. Of these, 76 have side effect information, and 58 are specifically associated with the four serious side effects mentioned above. The association network is shown in Figure 6A. Furthermore, the three new targets with the smallest *p*-values (STK17A, STK17B, and PRKAA2) are all connected to these side effects, reinforcing the effectiveness of target repositioning using pocket similarity to identify the proteins associated with the side effects of a drug. Appendix A shows that all 76 newly identified targets with side effect information are related to new side effects, providing novel insights into the drug safety of axitinib.

The analysis of sorafenib yielded similar results. As shown in Table 1, sorafenib has 130 known targets and 4 known pockets. Sorafenib was repositioned to 107 targets, with 22 overlaps with known targets. Of the 68 documented side effects of sorafenib, 60 are associated with these newly predicted targets, particularly the serious side effect of skin disease. Of the 85 new potential targets identified, 63 have associated side effects and 61 are related to adverse skin disease effects. The association network for sorafenib is shown in Figure 6B, and Appendix A indicates that all 63 newly identified targets with side effect information are linked to novel side effects, shedding light on the side effects of sorafenib.

The drug–protein–side effect network is a valuable tool for elucidating the nature of the side effects of drugs. These findings could significantly improve drug safety evaluation and facilitate the development of new therapeutic indications for existing drugs.

## 3. Discussion

Target identification is an important step in drug development [5]. Currently, research is focused on a limited number of targets; however, many targets have not been fully explored or have not been explored at all [35]. We aimed to assess the druggable potential of the human proteome by exploring the predictability of the druggability of pockets. Using protein structures predicted by AlphaFold2 and computational pocket prediction tools, including Fpocket, PUResNetV2, and P2Rank, this study provides a comprehensive analysis of the druggable landscape of the human proteome. Our analysis shows that the number of predicted druggable proteins in the human proteome is significantly higher than the 3000 proteins estimated by Hopkins [11]. Specifically, we identified 11,378 druggable proteins and 15,043 druggable pockets using Fpocket, demonstrating the vast untapped potential within the human proteome. Notably, even less-characterized or unstudied Tdark proteins exhibited druggable proportions exceeding 50%, challenging the traditional focus on well-characterized proteins (Tclin and Tchem). This highlights the importance of Tdark proteins as valuable targets for drug discovery and development. By ranking Tdark proteins based on their druggability, we identified high-priority candidates across various protein families, including ion channels, GPCRs, and enzymes, suggesting significant potential for novel drug development. These findings hold great promise for future drug development and side effect research.

By comparing the similarity of drug pockets across protein families, we revealed substantial interfamily pocket similarities, with notable interactions observed among the KRAB, PRY, and Ion_trans families. These findings highlight the structural versatility of protein families like KRAB, which can form similar pockets with multiple other families. This makes them attractive candidates for drug repurposing. Furthermore, the pocket similarity between diverse protein families suggests that drugs targeting specific families could potentially be repositioned to act on other families, thereby expanding their therapeutic applications. Based on the pocket similarity, we have successfully repositioned drugs and identified new therapeutic indications and side effects for existing drugs. For instance, we repositioned progesterone, estradiol, and alprazolam to interact with new protein targets, revealing potential therapeutic applications for conditions such as pemphigus, shingles, and Pick’s disease. Importantly, the drug–protein–side effect network we developed provides a new perspective on drug safety by associating repositioned targets with potential side effects. For example, axitinib and sorafenib were linked to previously uncharacterized side effects, such as hypertension and skin diseases, through newly predicted targets. This highlights the utility of our approach in drug repurposing and enhancing drug safety assessments.

Fpocket was chosen as the primary tool for predicting druggable pockets in this study due to its widespread use and demonstrated reliability in related research. A recent study has shown that Fpocket exhibits the highest recall and coverage, exceeding 99% among the 13 mainstream methods [36]. In terms of parameter selection, a lower threshold may result in more pockets predicted, but may also increase the number of false positives. We chose a strict drug score threshold of greater than 0.5 to ensure that the drug pockets were reliable and to guarantee the accuracy of subsequent studies. We predicted 15,043 druggable pockets from 11,378 druggable proteins, with a proportion of druggable proteins exceeding 50%. The predicted number of proteins and pockets would be different if pockets were defined using different thresholds. To further validate our results, we compared Fpocket’s predictions with those from PUResNetV2, a method shown to have the highest F1 score in recent studies. Our comparison revealed that Fpocket and PUResNetV2 predicted over 7000 overlapping druggable proteins (Appendix A). We also compared the proportion of druggable proteins predicted by Fpocket and PUResNetV2 for each family, as shown in Appendix A. In the GPCR, transporter, ion channel, and nuclear receptor families, Fpocket predicted a higher proportion of druggable proteins than PUResNetV2. In the enzyme, TF, kinase, and epigenetic family, Fpocket predicted a lower proportion of druggable proteins than PUResNetV2. These differences emphasize the importance of combining multiple tools to achieve a more comprehensive understanding of druggability across protein families. While enzymes and kinases are traditionally considered druggable, our results suggest that not all proteins in these families are druggable, which warrants further investigation. It is important to note that all druggability predictions in this study are based on computational methods, and experimental validation is needed to confirm whether these proteins are indeed druggable.

Our method is to predict the pocket similarity by using Apoc and to find the potential target of the drug according to the pocket similarity. Once the targets are predicted, we infer new therapeutic indications and side effects by mapping these targets to known relationships with diseases and side effects, sourced from established databases such as DrugCentral, DrugBank, TTD, and SIDER. We provided a complete list of 220,312 pairs of similar pockets that exist among the identified druggable pockets in Appendix A and a list of 62,077 pairs of similar pockets that exist among identified druggable pockets and known drug pockets in Appendix A. It would be useful for drug compound design. To validate the advantages of our method in drug repositioning, we chose the latest drug target prediction method ReduMixDTI [37] to compare with our method. We evaluated the proportion of tested compounds whose any true target is correctly predicted within the top N ranked targets. Our method outperforms ReduMixDTI across all evaluated Top-N accuracy levels (Top 10, Top 30, Top 50, and Top 100) (Appendix A), indicating that our method has advantages in the target accuracy of high-ranking prediction. In the field of drug repositioning, it is important to be able to accurately identify the few most likely targets, which can be very valuable for saving resources and speeding up the drug development process.

However, the proposed method also has certain limitations. First, the drug repositioning strategy based on pocket similarity is only applicable to drugs with known pocket structures. Of the 19,443 protein–ligand complex structures in the Pdbbind database [38], only 1872 molecules in the DrugBank database [39]. This limits the number of drugs that can be repositioned. Second, although the use of AlphaFold2-predicted structures for docking simulations may have limitations [40,41], we chose to use AlphaFold2-predicted structures to guarantee the acquisition of structural data for all human proteins given the comprehensive scope of this study, encompassing the entire human proteome. In the future, more accurate protein structures should be included to improve the accuracy of pocket prediction. Due to the large number of pocket similarity comparisons, and because Apoc can be used to quickly compare pocket similarity, we used Apoc for the pocket comparison. [42]. We adopted a strict *p*-value (<0.0001) to define similar pocket pairs and perform subsequent analysis due to the excessive number of pocket pairs (hundreds of millions of pairs). Similar pockets may also differ if different software is used to compare pockets or if different thresholds are used to define similar pocket pairs.

In summary, by using AlphaFold2-predicted structures, we have comprehensively analyzed the druggability of the human proteome, successfully repositioned existing drugs, and established a detailed network linking drugs, known targets, repositioned targets, and their associated indications and potential side effects. This network has the potential to identify new therapeutic applications for existing drugs and provide valuable insights into drug safety and off-target effects, ultimately contributing to more effective and safer drug development.

## 4. Methods

### 4.1. Protein 3D Structure Acquisition

The human proteome proteins that were manually reviewed and annotated (and thus reviewed) from the UniProt database [43] were subsequently obtained, and their 3D structures (20,255 proteins and their structures) were determined using AlphaFold2 (v2.0).

### 4.2. Proteome Druggable Pocket Prediction

Fpocket was employed to predict the existence of pockets in the aforementioned proteins. Fpocket is a geometry-based program that uses a grid-based method to identify potential binding pockets on the surface of protein structures by analyzing their geometry. The output for each predicted pocket is a PDB file containing the spatial coordinates of the atoms that form the pocket, as well as relevant physicochemical properties. The output also includes a table with additional information such as drug score, hydrophobicity score, polarity score, amino acid-based volume score, and charge score. A drug score greater than 0.5 defines a pocket as druggable. Proteins with druggable pockets were thus designated as druggable proteins. The overall druggability of the human proteome was visualized. P2Rank was used to predict pockets. Pockets with a probability greater than 0.5 were considered druggable. PUResNetV2 was used to predict druggable pockets without threshold.

### 4.3. Proteome Information Retrieval

The structural domain information for the human proteome was obtained from the UniProt database. Information pertaining to the target development level (TDL) and protein family category was sourced from Pharos [44]. Pharos categorizes TDL into four classes: Tclin, Tchem, Tbio, and Tdark. Tclin proteins are drug targets linked to at least one approved drug. Tchem proteins are known to bind to small molecules with high potency. Tbio denotes proteins with experimental evidence and disease links but are not classified as drug targets or do not meet the Tchem/Tclin standards. Tdark refers to the remaining proteins that were manually curated at the primary sequence level in UniProt but do not meet any of the criteria for Tclin, Tchem, or Tbio. Information pertaining to protein families was collated from the Pfam [45] database. Data concerning potential adverse effects of pharmaceutical agents were obtained from the SIDER [46] database. Information regarding the relationship between drugs, targets, and the indications for which they are used was integrated from the DrugCentral [47], DrugBank, TTD [48], and Pharos databases.

### 4.4. Target–Small-Molecule Complex Structure Acquisition

The protein–ligand complex structure files were sourced from the Pdbbind database (version 2020), comprising 19,443 entries. The CID numbers of the ligands were ascertained using RDKit (v2022.9.5) and PubChemPy (v1.0.4). Based on these CIDs, 1214 small molecules were selected from the DrugBank database.

### 4.5. Pocket Similarity Comparison

The druggable pockets were globally compared within the human proteome, and the human proteome druggable pockets and drug molecule pockets were compared using Apoc. Apoc requires the input of the structure of the two pockets and the structure of the corresponding protein and compares the two pockets by considering the geometric shape, chemical properties, and structural similarity of the pockets. The prediction results obtain the pocket similarity score (PS-score), *p*-value, RMSD, and matching residue list of the two pockets. Pocket pairs with a *p*-value < 0.0001 were defined as similar pocket pairs. The selections were further refined based on target family information and structural domain information, which was followed by visualization using Cytoscape (v3.10.2) [49].

### 4.6. Drug Repositioning Based on Pocket Similarity

A drug–known target–repositioned target network was constructed in which drug molecules were repositioned to other targets within the human proteome on the basis of pocket similarity. Molecular docking between drugs and repositioned targets was performed using AutoDock Vina (v1.1.2) [50], with visualization performed using PyMOL (v2.5.0).

### 4.7. Prediction of Drug Side Effects Based on Pocket Similarity

A drug–target–side effect network was constructed to identify repositioned targets based on pocket similarity. The set of repositioned targets, when subtracted from the known drug targets, yielded the drug’s off-target targets. The side effects of the drugs were interpreted based on the association between drug off-target targets and side effects. This approach allows for the prediction of potential side effects, thereby enhancing our understanding of a drug’s safety profile.

## Figures and Tables

**Figure 1 molecules-30-00260-f001:**
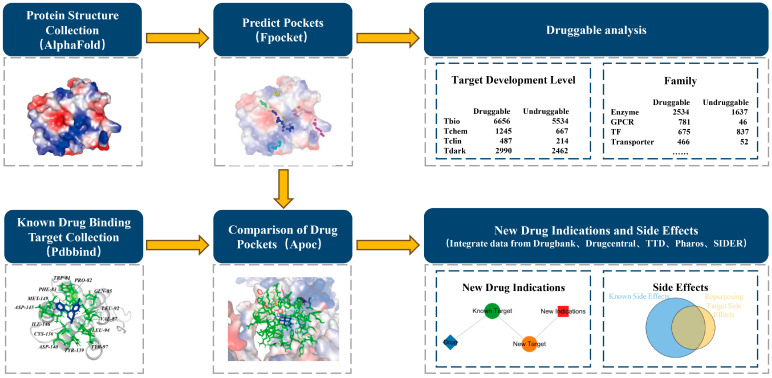
Flow chart of drug analysis and comparison of drug pockets. Protein prediction structures were obtained, and the druggability of the human proteome was analyzed through the prediction of pockets. The predicted pockets were then compared with those of known drug molecules, and the drug was repositioned according to the similarity of the pockets. This resulted in the discovery of new indications and side effects of the drug.

**Figure 2 molecules-30-00260-f002:**
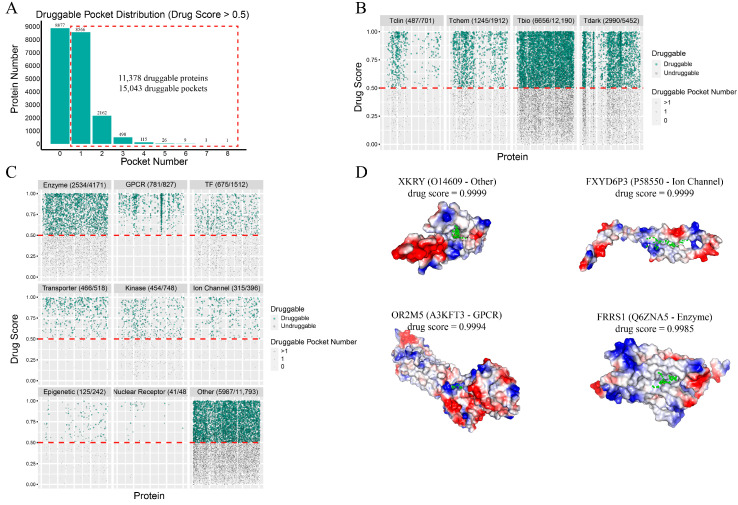
Druggability analysis of human proteome. (**A**) Bar chart of druggable pockets and druggable protein quantities in the human proteome predicted using Fpocket; 15,043 druggable pockets and 11,378 druggable proteins were identified. (**B**) Druggability in the human proteome at the target development level. The predicted fraction of druggable proteins exceeded 50% in each of the categories. Proteins above the red dashed line are druggable proteins. (**C**) Druggability in the human proteome at the family level. Proteins above the red dashed line are druggable proteins. (**D**) Examples of pockets of Tdark proteins with the highest drug scores from different families; green circles represent pocket (not a specific ligand). Name, family, and drug score of each protein are provided above the structure.

**Figure 3 molecules-30-00260-f003:**
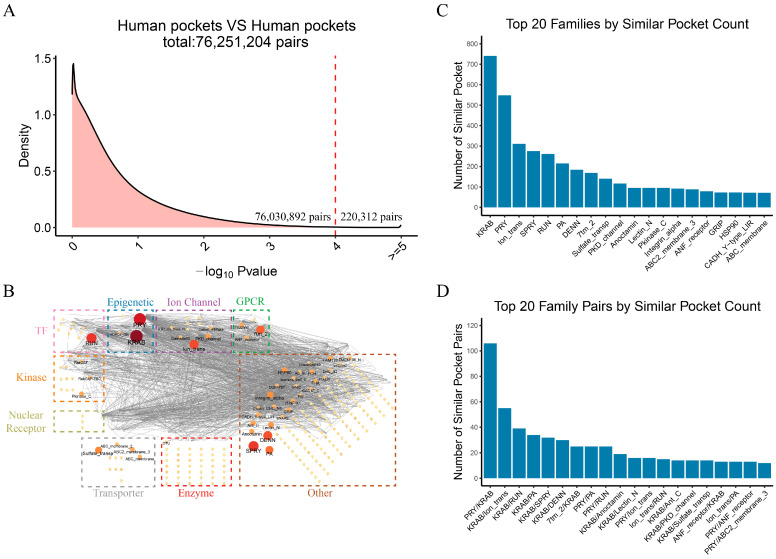
Global comparison of human proteome druggable pockets. (**A**) Density plot of druggable pocket similarity comparisons in the human proteome. Pocket pairs with a similarity defined by −log_10_
*p*-value > 4 are considered similar pocket pairs; 220,312 similar pocket pairs were identified. the red dash line indicates the boundary between similar and dissimilar pocket pairs. (**B**) Network of similar pocket pair families. Node represents protein family; the larger the node size and the redder the color, the greater the number of similar pocket pairs formed with other families, indicating that they are more likely to be repositioned by small molecules. (**C**) Bar chart of the top 20 families with the highest number of similar pocket pairs formed with all other families. Vertical axis represents the number of similar pocket pairs formed with all other family pockets. The KRAB, PRY, and Ion_trans families are in the top three for the number of similar pocket pairs formed with other families. (**D**) Bar chart of the top 20 family pairs with the highest number of similar pocket pairs. Family pairs are named with human protein families listed first, followed by families of known drug targets. Vertical axis represents the number of similar pocket pairs between the paired families. The PRY/KRAB family has the highest number of similar pocket pairs.

**Figure 4 molecules-30-00260-f004:**
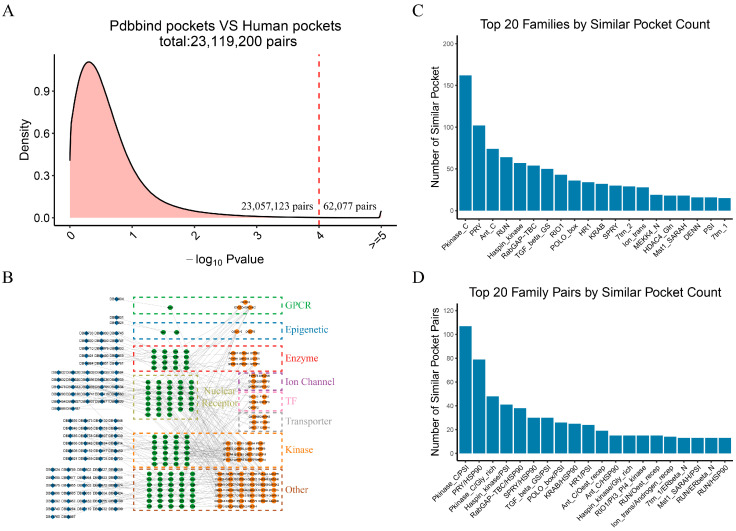
Comparison of human proteome druggable pockets and small-molecule drug pockets. (**A**) Density plot of druggable pocket similarity comparisons between the human proteome and small-molecule drug pockets. Pocket pairs with a similarity defined by −log_10_
*p*-value > 4 are considered similar pocket pairs; 62,077 similar pocket pairs were identified. the red dash line indicates the boundary between similar and dissimilar pocket pairs. (**B**) Network of drug, known target, repositioned target. Blue diamonds, green circles, and orange circles represent drugs, known targets, and repositioned targets, respectively. (**C**) Bar chart of the top 20 families with the highest number of similar pocket pairs formed with all drug pockets in the human proteome. The Pkinase_C, PRY, and Ant_C families are in the top three for the number of similar pocket pairs formed with other families. (**D**) Bar chart of the top 20 family pairs with the highest number of similar pocket pairs. Family pairs are named with human protein families listed first, followed by families of known drug targets. Vertical axis represents the number of similar pocket pairs between the paired families. The Pkinase_C/PSI family has the highest number of similar pocket pairs.

**Figure 5 molecules-30-00260-f005:**
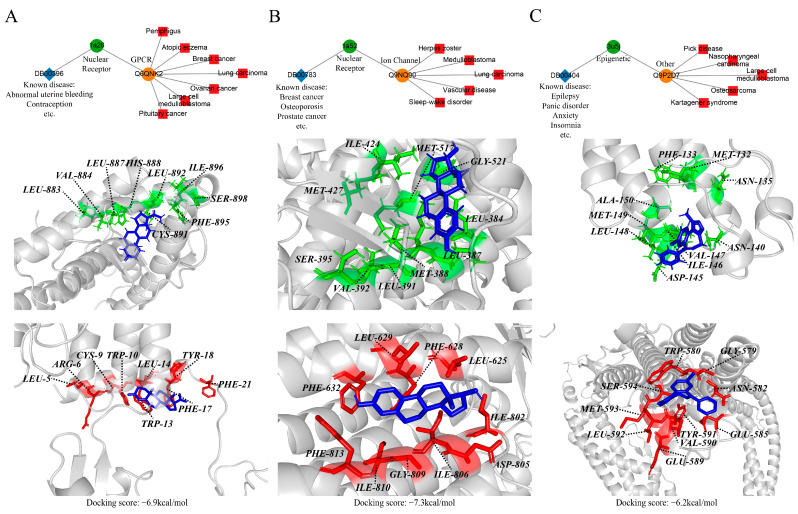
Examples of drug repurposing. The landscape of drug repositioning for (**A**) progesterone, (**B**) estradiol, and (**C**) alprazolam was shown. DB00396 is repositioned to GPCR target Q6QNK2 by known Nuclear Receptor target (pdbid: 1a28). DB00783 is repurposed to ion channel target Q9QN90 by known Nuclear Receptor target (pdbid: 1a52). DB00404 is repositioned to other target Q9P2D7 by known epigenetic target (pdbid: 3u5j). Upper panel: The drug repositioning process; blue diamonds, green circles, orange circles, and red squares represent drugs, known targets, repositioned targets, and new indications, respectively. Middle panel: Close view of drug binding to known pockets. Lower panel: Close view of drug binding to repositioned pockets. The residues depicted in the figure are the corresponding residues from the similar pocket pairs predicted by Apoc.

**Figure 6 molecules-30-00260-f006:**
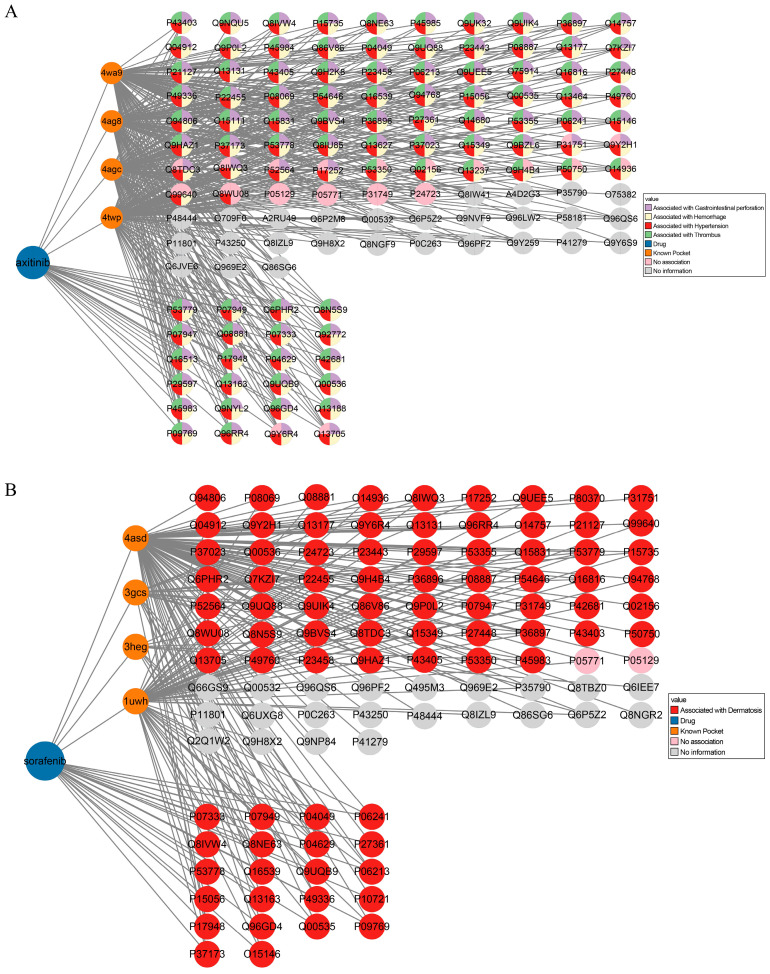
Drug–known target–repositioned target side effect association network. The landscape of (**A**) axitinib as well as (**B**) sorafenib and estradiol repositioning was shown. Axitinib is repurposed to 127 targets (24 known targets and 103 unknown targets) by 4 known targets. Here, 76 unknown targets were associated with at least one drug side effect. Sorafenib is repurposed to 107 targets (22 known targets and 85 unknown targets) by 4 known targets. Additionally, 61 unknown targets were associated with drug side effect. Blue circles, orange circles, and other circles represent drug, known targets, and repositioned targets, respectively.

**Table 1 molecules-30-00260-t001:** Drug repurposing and side effect intersection analysis.

Drug Name	Number of Drug Side Effects ^1^	Number of Known Targets ^2^	Number of Predicted Known Targets ^3^	Numer of Overlapping Targets ^4^	Number of Predicted Unknown Targets ^5^	Number of Overlap of Known and Predicted Side Effects ^6^
ribavirin	242	8	4	0	4	0
nilotinib	144	75	89	9	80	102
topiramate	74	47	6	2	4	0
alprazolam	71	11	7	0	7	0
dabrafenib	71	9	113	3	110	64
sorafenib	68	130	107	22	85	60
gemcitabine	62	8	9	1	8	8
bortezomib	57	36	1	0	1	0
estradiol	48	22	214	3	211	33
axitinib	48	99	127	24	103	46
tolcapone	47	6	15	1	14	0
vemurafenib	43	8	109	3	106	35
acyclovir	28	7	1	0	1	0
entacapone	27	4	15	1	14	0
efavirenz	27	6	5	0	5	0
eplerenone	23	4	89	0	89	13
mifepristone	23	22	96	1	95	19
levonorgestrel	22	7	12	1	11	0
nevirapine	21	5	4	0	4	0
theophylline	21	20	1	0	1	0
testosterone	18	13	106	0	106	8
progesterone	17	30	134	1	133	15
apixaban	16	3	4	1	3	0
celecoxib	14	61	3	2	1	0
thymidine	13	1	20	0	20	0
tadalafil	11	18	67	5	62	9
adenosine	10	21	235	5	230	9
trimetrexate	10	13	1	0	1	0
trimethoprim	8	26	15	0	15	7
dexamethasone	6	13	125	0	125	4
ciclopirox	5	2	2	0	2	0
colchicine	4	15	14	1	13	4
dolutegravir	4	6	1	0	1	0
valdecoxib	4	37	2	2	0	0
etravirine	3	10	5	0	5	0
rivaroxaban	2	3	4	0	4	2
pentoxifylline	2	4	66	2	64	2
ticagrelor	1	3	5	0	5	0
flumazenil	1	38	2	0	2	0

^1^ The number of known side effects of the drug. ^2^ The number of known targets of the drug. ^3^ The number of predicted known targets according to pocket similarity. ^4^ The number of overlapped known targets and predicted targets. ^5^ The number of predicted targets that are not in the known target. ^6^ The number of overlapped known and predicted side effects.

## Data Availability

The original data and Appendix A presented in the study are openly available in FigShare and accessed at the following DOI: https://doi.org/10.6084/m9.figshare.27894594.

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
