# Peer review of "Proteome-Wide Identification and Comparison of Drug Pockets for Discovering New Drug Indications and Side Effects"

_molecules, 2025, doi:10.3390/molecules30020260_

Round 1
Reviewer 1 Report
Comments and Suggestions for Authors
In this work, a great deal of work has been done to analyse the structures of human proteome proteins, to identify potential binding sites in them by FPocket, to compare these sites in different proteins, to compare them with known binding sites of small chemical compounds and, thus, to identify unknown molecular targets of action of existing drugs to explain the reason of side effects. Undoubtedly, this is a very important and relevant task. However, the design of the article has serious flaws.
1. It would be useful to explain in the Methods exactly what information you obtained at each stage of the research. It would be worth describing very briefly what exactly the FPocket programme does to identify potential pockets in the protein structure and in what form it outputs the result. I assume that each predicted pocket consists of a list of amino acid residues extending into the cavity of that pocket. Accordingly, to compare two pockets, two protein structures and two lists of involved amino acid residues must be fed to Apoc's input, respectively. Apoc best matches the listed amino acid residues in space, then looks at the proximity of their physicochemical properties. If my assumptions are correct, it is necessary to supplement the Methods with relevant information.
2. References to the tools, such as Fpocket, P2Rank, Apoc and others need to be put at the place of first occurrence in the text, apart from the Methods.
3. In section 2.1. ( line 117) it is not clear what Tclin, Tbio, Tdark, Tche are. The information is given in the Methods, but the Methods are located much later in the text. Add literally a couple of phrases for clarification, so that there is no need to stop reading and go to the Methods.
4. Some figures have inserts of rather poor quality or it is impossible to read too small font, e.g. Fig.2, 4, 5 (top panel), S3(right panel), S4 (right panel).
5. Captions to figures and tables do not give a complete picture of what they depict. For example, Figure 2.S. - what is depicted in the figure? Figure 2D - what are the ligands located at the protein binding sites? Figure 3C, D - what does ‘number of similar pockets’ mean? How is this value calculated? Fig. 3B. - it is not clear what is depicted. In the caption to figure 3, 4 and everywhere else in the text in the formula ‘log10P-value’ 10 should be written by subscript. In the caption for Figures 3 and 4, the minus is missing and the comparison sign is mixed up. It is necessary to rewrite all figure captions so that it is clear what is shown in the figures without additional information.
6. Tables 1 and 2 look incomprehensible and meaningless, the captions to them also leave much to be desired. In Table 3, it is not clear how the numbers in the cells are obtained and what is the difference between ‘repurposing’ and ‘new’ target. Tables S1-S3 are unclear as to what they contain - their captions are completely inexplicable. In general, I have the impression that all the tables except for the table S5 can be deleted without loss of information. Besides, the captions to the tables should go before the table, not after it.
7. Where can I find a complete list of 220,312 similar binding sites ? I would like to manually compare a couple of such sites to evaluate the quality of Apoc. Also, such a list would be useful in drug compound design.
Comments on the Quality of English Language
The article contains too many typos, I advise you to ask an English-speaking colleague to proofread your text.
Author Response
Comments 1: In this work, a great deal of work has been done to analyse the structures of human proteome proteins, to identify potential binding sites in them by FPocket, to compare these sites in different proteins, to compare them with known binding sites of small chemical compounds and, thus, to identify unknown molecular targets of action of existing drugs to explain the reason of side effects. Undoubtedly, this is a very important and relevant task. However, the design of the article has serious flaws.
Response 1: We thank Reviewer #1 for his thoughtful comments and the time and effort he took to evaluate our manuscript. We appreciate the recognition of the importance and relevance of our work. Comments and suggestions are helpful in improving our manuscript. We have carefully revised the manuscript based on the comments and the detailed responses are as follows.
Comment 2: It would be useful to explain in the Methods exactly what information you obtained at each stage of the research. It would be worth describing very briefly what exactly the FPocket programme does to identify potential pockets in the protein structure and in what form it outputs the result. I assume that each predicted pocket consists of a list of amino acid residues extending into the cavity of that pocket. Accordingly, to compare two pockets, two protein structures and two lists of involved amino acid residues must be fed to Apoc's input, respectively. Apoc best matches the listed amino acid residues in space, then looks at the proximity of their physicochemical properties. If my assumptions are correct, it is necessary to supplement the Methods with relevant information.
Response 2: Thank you for your valuable comments and suggestions. As suggested, we have updated the manuscript to describe Fpocket in more detail. Specifically, we have added the following revised description: “Fpocket is a geometry-based program that uses a grid-based method to identify potential binding pockets on the surface of protein structures by analyzing their geometry. The output for each predicted pocket is a PDB file containing the spatial coordinates of the atoms that form the pocket, as well as relevant physicochemical properties. The output also includes a table with additional information such as Drug Score, Hydrophobicity Score, Polarity Score, Amino Acid-based Volume Score, and Charge Score.”(lines 433-438).
We have also revised the manuscript to include a clearer description of how the Apoc tool compares pockets. “Apoc requires as input the structure of the two pockets to be compared, along with the corresponding protein structures. It compares these pockets by evaluating their geometric shape, chemical properties, and structural similarity. The comparison results in several output metrics, including the Pocket Similarity Score (PS-score), P-value, RMSD (Root Mean Square Deviation), and a list of the matching residues between the two pockets.”(lines 466-470)
Comment 3: References to the tools, such as Fpocket, P2Rank, Apoc and others need to be put at the place of first occurrence in the text, apart from the Methods.
Response 3: We appreciate the reviewer's attention to this detail. We have adjusted the position of the references throughout. (lines 498-629).
Comment 4: In section 2.1. ( line 117) it is not clear what Tclin, Tbio, Tdark, Tchem are. The information is given in the Methods, but the Methods are located much later in the text. Add literally a couple of phrases for clarification, so that there is no need to stop reading and go to the Methods.
Response 4: Thank you very much for your valuable comments. We have added a brief description of each category in the revised manuscript as follows. “Tclin refers to druggable targets with approved drugs. Tchem are proteins that are not Tclin but are known to bind small molecules with high potency. Tbio are proteins with disease associations but no small molecule binding. Tdark are proteins that do not meet Tclin, Tchem or Tbio criteria (see Methods for detailed definitions).”(lines 114-117).
Comment 5: Some figures have inserts of rather poor quality or it is impossible to read too small font, e.g. Fig.2, 4, 5 (top panel), S3(right panel), S4 (right panel).
Response 5: Thank you for your valuable comments. As suggested, we have increased the font size and improved the overall quality of the images to ensure better visibility and clarity.
Comment 6: Captions to figures and tables do not give a complete picture of what they depict. For example, Figure 2.S. - what is depicted in the figure? Figure 2D - what are the ligands located at the protein binding sites? Figure 3C, D - what does ‘number of similar pockets’ mean? How is this value calculated? Fig. 3B. - it is not clear what is depicted. In the caption to figure 3, 4 and everywhere else in the text in the formula ‘log10P-value’ 10 should be written by subscript. In the caption for Figures 3 and 4, the minus is missing and the comparison sign is mixed up. It is necessary to rewrite all figure captions so that it is clear what is shown in the figures without additional information.
Response 6: We sincerely appreciate the reviewer’s detailed comments on the figure captions. We agree that clear and comprehensive captions are essential for readers to fully understand the content of the figures. As mentioned above, we have made the following revisions:
We have revised the captions for Figure 2S, Figure 2D, Figure 3C, 3D, and Figure 3B to more clearly describe what is depicted in each figure.
For Figure 2S, we have added a description of the content being shown. “The number of druggable proteins predicted by Fpocket, P2Rank and PUResNetV2. (A) The number of druggable proteins predicted by Fpocket, P2Rank and PUResNetV2 without screen, all the druggable proteins predicted by PUResNetV2 can be predicted by Fpocket. (B) The number of druggable proteins predicted by Fpocket, P2Rank, and PUResNetV2 after screening, Fpocket and PUResNetV2 predicted more than 7,000 druggable protein intersections. (C) The number of pockets predicted by PUResNetV2 intersecting with those predicted by Fpocket.”
For Figure 2D, we have clarified what the ligands are and their location at the protein binding sites. The green circle in the figure is not a specific ligand, but represents the spatial position occupied by the pocket.
For Figures 3C and 3D, we have defined what is meant by the "number of similar pockets" and explained how this value is calculated. The "Number of similar pockets" is the number of similar pocket pairs that a family's pockets form with all other family's pockets based on pocket similarity. The "Number of similar pocket pairs" is the number of similar pocket pairs between the paired families based on pocket similarity.
For Figure 3B, we have provided a clear explanation of what is depicted in the figure.
We have corrected the formatting of the "log10 P-value" throughout the manuscript and captions, ensuring that the "10" is in subscript as required.
We have also addressed the formatting issues with the minus sign and comparison symbol in the captions for Figures 3 and 4, ensuring they are correctly represented.
Comment 7: Tables 1 and 2 look incomprehensible and meaningless, the captions to them also leave much to be desired. In Table 3, it is not clear how the numbers in the cells are obtained and what is the difference between ‘repurposing’ and ‘new’ target. Tables S1-S3 are unclear as to what they contain - their captions are completely inexplicable. In general, I have the impression that all the tables except for the table S5 can be deleted without loss of information.
Response 7: Thank you for your valuable comment.
The original intent of Tables 1 and 2 was to show the identification of similar pocket pairs across different protein families and domains. However, we realize that the presentation was not effective in conveying the intended message. Therefore, as your suggestion, we have removed Tables 1 and 2 from the manuscript. We have provided a more detailed explanation of the process for identifying similar pocket pairs in the revised manuscript. (lines 162-165).
We have updated Table 3 to clarify the methodology used to derive the numbers in the cells. Specifically, we have renamed the columns as "Number of predicted known targets" and "Number of predicted unknown targets". Predicted known targets refer to proteins that are already targeted by a given drug. Predicted unknown targets are proteins identified by our analysis that are not known targets of a given drug. In addition, we have revised the caption to make these distinctions clearer and facilitate better understanding. (lines 324-327).
Table S1 (now Table S2) is intended to show the twenty Tdark proteins with the highest Drug Score, as these proteins represent an underexplored but potentially valuable source of new therapeutic targets. We have revised the caption to better explain the significance of studying these high-scoring Tdark proteins, which could lead to the discovery of novel drug targets.
After carefully reconsidering these tables, we agree with the reviewer’s assessment. Tables S2, S3, and S4 were found to provide limited additional information and have been deleted in the revised manuscript.
Comment 8: Where can I find a complete list of 220,312 similar binding sites? I would like to manually compare a couple of such sites to evaluate the quality of Apoc. Also, such a list would be useful in drug compound design.
Response 8: Thank you for your valuable comment. Following your suggestion, we have added a complete list of 220,312 pairs of similar pockets among all identified druggable pockets (Table S4). Additionally, we have added a list of 62,077 pairs of similar pockets among the identified druggable pockets and known drug pockets (Table S5). To facilitate your manual comparison and evaluation of the quality of Apoc, the two lists have been uploaded to Figshare and can be accessed at the following DOI: dx.doi.org/10.6084/m9.figshare.27894594.
Comment 9: The article contains too many typos, I advise you to ask an English-speaking colleague to proofread your text.
Response 9: Thank you for your advice. After all revisions, our manuscript has undergone English language editing by MDPI Author Services. The text has been checked for correct use of grammar and common technical terms, and edited to a level suitable for reporting research in a scholarly journal.
Reviewer 2 Report
Comments and Suggestions for Authors
This study leveraged the Fpocket tool to assess the druggability of the entire human proteome, identifying 15,043 druggable pockets and significantly expanding the druggable proteome. The research also demonstrated how pocket similarity can be applied to drug repurposing, target extension, and drug safety evaluation, presenting new avenues for identifying novel drug indications and side effects of existing drugs. However, several aspects of the research design warrant further attention, and the following comments may help improve the manuscript:
1. Why were Fpocket and P2Rank selected for this study? What is the accuracy of their predictions? Fpocket, introduced in 2009, and P2Rank, published in 2018, are relatively dated. Are there more recent or more accurate tools available for such predictions? If the prediction accuracy of these methods is insufficient, discussing the number of druggable target proteins predicted becomes less meaningful. Furthermore, the selection of parameters has a direct impact on the number of predicted druggable targets, and thus the significance of this metric should be reconsidered.
2. The study reports that “The GPCR, transporter, and nuclear receptor families showed the highest druggable proportions with 94.44%, 89.96%, and 85.42%, respectively.” This result seems at odds with common knowledge, as the proportion of druggable proteins is typically thought to be much lower. Could this discrepancy be a result of the inherent limitations in Fpocket’s prediction accuracy?
3. The molecular docking results provided cannot conclusively validate the drug repurposing predictions, as molecular docking primarily predicts the binding potential of drug molecules to proteins, rather than confirming their therapeutic efficacy. Moreover, when validating drug-disease predictions, it is insufficient to present just a few examples (e.g., progesterone, estradiol, alprazolam). A more comprehensive comparison with other state-of-the-art (SOTA) methods across multiple benchmark datasets is needed to demonstrate the advantages of the drug repurposing approach based on binding pocket similarity.
4. Similarly, the prediction of drug side effects should also be compared with other state-of-the-art (SOTA) methods, rather than relying solely on statistical tables (e.g., Table 3) to demonstrate the advantages and correctness of the approach.
5. The discussion section lacks depth, as it primarily addresses the limitations of the current study without offering a more comprehensive analysis of the results or their broader implications.
Author Response
Comment 1: This study leveraged the Fpocket tool to assess the druggability of the entire human proteome, identifying 15,043 druggable pockets and significantly expanding the druggable proteome. The research also demonstrated how pocket similarity can be applied to drug repurposing, target extension, and drug safety evaluation, presenting new avenues for identifying novel drug indications and side effects of existing drugs. However, several aspects of the research design warrant further attention, and the following comments may help improve the manuscript:
Response 1: We thank Reviewer #2 for his thoughtful comments and the time and effort he took to evaluate our manuscript. We appreciate the recognition of the importance and relevance of our work. The comments and suggestions were very helpful in improving the manuscript. We have revised the manuscript based on the comments, and below are our detailed responses to the comments.
Comment 2: Why were Fpocket and P2Rank selected for this study? What is the accuracy of their predictions? Fpocket, introduced in 2009, and P2Rank, published in 2018, are relatively dated. Are there more recent or more accurate tools available for such predictions? If the prediction accuracy of these methods is insufficient, discussing the number of druggable target proteins predicted becomes less meaningful. Furthermore, the selection of parameters has a direct impact on the number of predicted druggable targets, and thus the significance of this metric should be reconsidered.
Response 2: We sincerely thank the reviewer for the insightful comments.
Fpocket was selected as the primary tool due to its widespread use and demonstrated reliability in relevant studies. According to the most recent study (Utgés JS, Journal of Cheminformatics, 2024), Fpocket has the highest recall and coverage, with both metrics exceeding 99%. This indicates that Fpocket can predict nearly all relevant pockets on the surface of proteins and accurately identify the most significant ones. In addition, Fpocket provides a drug score for each predicted pocket, ranging from 0 to 1. This score serves as a metric for the likelihood of a pocket binding drug-like molecules, making it an effective tool for screening and prioritizing druggable pockets.
We acknowledge the reviewer's concern regarding the use of relatively older tools like Fpocket (introduced in 2009) and P2Rank (2018). While these tools may seem dated, they remain highly effective for predicting druggable pockets. Recent comparisons have confirmed that Fpocket remains one of the most accurate tools available. In fact, its high recall and coverage rates indicate that it is capable of identifying almost all relevant pockets with high confidence. To further validate the accuracy of Fpocket's predictions, we compared its results with those of PUResNetV2, a more recent tool with strong performance metrics (Jeevan K, Journal of Cheminformatics, 2024). Our comparison showed that all druggable proteins predicted by PUResNetV2 could also be predicted by Fpocket (Figure S2A). When a strict drug score threshold (>0.5) was applied, we found that Fpocket and PUResNetV2 predicted over 7,000 overlapping druggable protein intersections (Figure S2B), further supporting the reliability of Fpocket's predictions. In addition, of the 14,450 pockets predicted by PUResNetV2, 13,493 pockets (93.38%) were overlapped with those predicted by Fpocket (Figure S2C), confirming that Fpocket remains highly effective even when compared to newer methods.
Regarding the impact of parameter selection, we fully recognize that parameter selection plays a critical role in determining the number of predicted druggable targets. To address this, we chose a strict drug score threshold of 0.5 to balance sensitivity and specificity. This threshold ensures that only pockets with a high probability of being druggable are selected, while minimizing the risk of false positives. While a lower threshold would increase the number of predicted pockets, it could also introduce less relevant or unreliable pockets. Thus, the 0.5 threshold prioritizes accuracy and reliability, which are critical to the robustness of our results.
We have added the above content in the revised manuscript. (lines 104-109, lines 364-387)
Comment 3: The study reports that “The GPCR, transporter, and nuclear receptor families showed the highest druggable proportions with 94.44%, 89.96%, and 85.42%, respectively.” This result seems at odds with common knowledge, as the proportion of druggable proteins is typically thought to be much lower. Could this discrepancy be a result of the inherent limitations in Fpocket’s prediction accuracy?
Response 3: Thank you for your comment.
The apparent discrepancy you noted regarding the high druggable proportions is related to the distinction between known druggable targets and predicted druggable targets. In our study, we used the concept of target development levels as defined in the Pharos database (https://pharos.nih.gov/), which categorizes targets into four levels: Tclin, Tchem, Tbio, and Tdark. Tclin refers to druggable targets with approved drugs. Tchem are proteins that are not Tclin but are known to bind small molecules with high potency. Tbio are proteins with disease associations but no small molecule binding. Tdark are proteins that do not meet Tclin, Tchem or Tbio criteria (see Methods for detailed definitions). According to the Pharos database, there are 20,412 total targets, with 704 Tclin and 1,904 Tchem targets. Among GPCRs, there are 827 total targets, of which 102 are Tclin and 160 are Tchem. The proportion of known druggable targets (Tclin and Tchem) is 13% (2608/20412) for all and 32% (262/827) for GPCR. This proportion is consistent with common knowledge and your comments.
However, our study focused on predicted druggable targets, which are identified by Fpocket’s druggable pocket predictions. These predicted targets expand the druggable proteome by including pockets that have the potential to bind small molecules, even if they are not currently considered druggable based on existing data. Therefore, the higher druggable proportions observed in our study reflect the predicted druggable targets, not just those with known drug associations.
We further compared the results of Fpocket with those of PUResNetV2, a more recent prediction tool. In families such as GPCRs, transporters, ion channels, and nuclear receptors, Fpocket predicted a higher proportion of druggable proteins compared to PUResNetV2. Conversely, in enzyme, transcription factor (TF), kinase, and epigenetic families, PUResNetV2 predicted a higher proportion of druggable proteins. In general, however, the differences between the two methods were minimal (Table S1).
In summary, the higher druggable proportions observed in our study are due to the use of predicted druggable targets, which expand upon the known druggable targets in the Pharos database. We hope this explanation clarifies the observed discrepancy and provides a better understanding of the study's methodology. Thank you again for your valuable comment.
We have added the above content in the revised manuscript. (lines 364-387)
Comment 4: The molecular docking results provided cannot conclusively validate the drug repurposing predictions, as molecular docking primarily predicts the binding potential of drug molecules to proteins, rather than confirming their therapeutic efficacy. Moreover, when validating drug-disease predictions, it is insufficient to present just a few examples (e.g., progesterone, estradiol, alprazolam). A more comprehensive comparison with other state-of-the-art (SOTA) methods across multiple benchmark datasets is needed to demonstrate the advantages of the drug repurposing approach based on binding pocket similarity.
Response 4: Thank you for your valuable comment.
We fully understand that molecular docking primarily assesses the binding potential of drug molecules to proteins, rather than directly confirming their therapeutic efficacy. We agree that drug repurposing predictions require a more comprehensive validation approach, beyond just a few examples. In our study, we focused on predicting potential drug targets based on pocket similarity using the Apoc tool. The primary goal of our method is to identify new therapeutic indications and side effects by mapping these predicted targets to known relationships between targets and indications, as well as targets and side effects, sourced from established databases such as DrugCentral, DrugBank, TTD, and SIDER. It is important to note that our approach does not aim to directly validate therapeutic efficacy through docking simulations; instead, it predicts potential targets based on the structural similarity of binding pockets.
To demonstrate the effectiveness of our drug repurposing approach, we focused on target prediction accuracy as the key performance metric. We compared our method with the ReduMixDTI model (Liu M, Journal of Cheminformatics, 2024), one of the latest drug-target prediction methods, using Top-N accuracy as the evaluation metric. This metric reflects the proportion of tested compounds whose any true target is correctly predicted within the top N ranked targets. In our analysis, we evaluated Top 10, Top 30, Top 50, and Top 100 accuracies for both the ChEMBL and BindingDB datasets.
Our results show that our method outperforms ReduMixDTI across all evaluated Top-N accuracy levels (Top 10, Top 30, Top 50, and Top 100), highlighting its advantage in predicting high-ranking drug targets (Table S6). In drug repurposing, the ability to accurately identify a small number of highly likely targets is crucial for resource efficiency and speeding up the drug development process. Since therapeutic indications are inferred from predicted target-indication relationships, the improved accuracy in target prediction further enhances the reliability of our indications predictions.
We have incorporated these updates into the revised manuscript. (lines 395-403)
Comment 5: Similarly, the prediction of drug side effects should also be compared with other state-of-the-art (SOTA) methods, rather than relying solely on statistical tables (e.g., Table 3) to demonstrate the advantages and correctness of the approach.
Response 5: Thank you for your comment. In our study, we used pocket similarity predictions via the Apoc tool to identify potential drug targets. We then mapped these targets to known target-disease and target-side effect relationships from established databases such as DrugCentral, DrugBank, TTD, and SIDER. This process allowed us to infer not only new therapeutic indications but also potential side effects.
To validate our method, we compared it with ReduMixDTI, a state-of-the-art drug-target prediction approach. In the response to comment 4, our results showed that our approach outperforms ReduMixDTI across all evaluated Top-N accuracy levels (Top 10, Top 30, Top 50, and Top 100), highlighting the strength of our method in predicting high-ranking drug targets (Table S6). Since side effects are inferred based on predicted target-side effects relationships, the improved accuracy in target prediction directly enhances the reliability of our side effect predictions.
We have incorporated this clarification and the comparison with ReduMixDTI into the revised manuscript to better demonstrate the advantages of our approach for both drug repurposing and side effect prediction. (lines 395-403)
Comment 6: The discussion section lacks depth, as it primarily addresses the limitations of the current study without offering a more comprehensive analysis of the results or their broader implications.
Response 6: Thank you for your valuable comment. We have revised and expanded the discussion to provide a more comprehensive analysis of our results and their broader implications (lines 328-425).
References
- Utgés JS, Barton GJ. Comparative evaluation of methods for the prediction of protein-ligand binding sites. Journal of Cheminformatics. 2024 Nov 11;16(1):126. doi: 10.1186/s13321-024-00923-z. PMID: 39529176; PMCID: PMC11552181.
- Liu M, Meng X, Mao Y, Li H, Liu J. ReduMixDTI: Prediction of Drug-Target Interaction with Feature Redundancy Reduction and Interpretable Attention Mechanism. Journal of Cheminformatics. 2024 Dec 9;64(23):8952-8962. doi: 10.1021/acs.jcim.4c01554. Epub 2024 Nov 21. PMID: 39570771.
- Jeevan K, Palistha S, Tayara H, et al. PUResNetV2. 0: a deep learning model leveraging sparse representation for improved ligand binding site prediction[J]. Journal of Cheminformatics, 2024, 16(1): 1-16.
Round 2
Reviewer 2 Report
Comments and Suggestions for Authors
All previously raised concerns have been addressed, and I endorse the acceptance of this paper for publication.